# Characterisation of Orthohantavirus Serotypes in Human Infections in Kazakhstan

**DOI:** 10.3390/v17070925

**Published:** 2025-06-28

**Authors:** Nur Tukhanova, Anna Shin, Abhishek Bakuli, Lyazzat Yeraliyeva, Nurbek Maikanov, Guenter Froeschl, Zauresh Zhumadilova, Gulnara Tokmurziyeva, Edith Wagner, Sandra Essbauer, Lukas Peintner

**Affiliations:** 1M. Aikimbayev’s National Scientific Center of Especially Dangerous Infections, 050054 Almaty, Kazakhstan; tukhanovanur@gmail.com (N.T.); zzbgdirect@nscedi.kz (Z.Z.); tokmurziyeva@yandex.ru (G.T.); 2Department of Infectious and Tropical Disease, S. Asfendiyarov Kazakh National Medical University, 050012 Almaty, Kazakhstan; 3Department of General Medical Practice, Al-Farabi Kazakh National University, 050040 Almaty, Kazakhstan; annashin86@gmail.com; 4Division of Infectious Diseases and Tropical Medicine, University Hospital, Ludwig-Maximilians-University, 80802 Munich, Germany; bakuli@lrz.uni-muenchen.de (A.B.);; 5National Scientific Center of Phthisiopulmonology, 050010 Almaty, Kazakhstan; l.eralieva@mail.ru; 6Oral Antiplague Station, 090001 Oral, Kazakhstan; nmaikanov@mail.ru; 7Institute of Medical Microbiology, Jena University Hospital, 07747 Jena, Germany; 8Department of Virology and Intracellular Agents, Bundeswehr Institute of Microbiology, 80937 Munich, Germany; 9Institute of Molecular Medicine and Cell Research, University of Freiburg, 79104 Freiburg im Breisgau, Germany

**Keywords:** Kazakhstan, orthohantavirus, serology, human infections

## Abstract

Orthohantavirus infection is a zoonotic disease transmitted to humans through contact with infected rodents. In Eurasia, Old World Orthohantaviruses can cause haemorrhagic fever with renal syndrome (HFRS), while in the Americas, New World Orthohantaviruses are responsible for hantavirus cardiopulmonary syndrome (HCPS). In Kazakhstan, the first recorded cases of HFRS appeared in the West Kazakhstan region in 2000, which has since then been established as an endemic area due to the presence of stable rodent reservoirs and recurring human infections. Routine diagnosis of HFRS in this region relies primarily on immunoassays. To enhance diagnostic precision, we aimed to implement both serological and molecular methods on samples from suspected HFRS cases in the endemic West Kazakhstan region and non-endemic Almaty City. A total of 139 paired serum, saliva, and urine samples were analysed using IgM/IgG ELISA, immunoblot assays, and qPCR. Our findings confirm that suspected HFRS cases in West Kazakhstan are associated with the Puumala virus serotype.

## 1. Introduction

The genus *Orthohantavirus* (family *Hantaviridae*) is geographically widespread and represents a pathogen of considerable public health relevance [1]. While several rodent species serve as natural reservoirs, orthohantaviruses have also been detected in shrews, moles, and bats [2]. These viruses are typically asymptomatic in their reservoir hosts; however, humans, considered as dead-end hosts, can develop severe disease manifestations. Persistently infected rodents continuously shed the virus through their excreta, with human infection occurring primarily via the inhalation of aerosolised, virus-contaminated excreta, and less commonly through direct contact with infected rodents. Two distinct clinical syndromes caused by rodent-borne orthohantavirus infections have been described: (i) haemorrhagic fever with renal syndrome (HFRS), predominantly in Eurasia, and (ii) hantavirus cardiopulmonary syndrome (HCPS), mainly observed in the Americas [3]. In European countries, the primary causative agents of HFRS are *Orthohantavirus puumalaense* (PUUV) and *Orthohantavirus dobravaense* (DOBV), whereas *Orthohantavirus hantanense* (HNTV) is the predominant driver of HFRS in Asia [4]. PUUV is known to cause Nephropathia epidemica (NE), a mild form of HFRS, while DOBV can lead to moderate to severe disease. To date, four genotypes of DOBV have been identified: Dobrava, Kurkino, Saaremaa, and Sochi. *Orthohantavirus seoulense* (SEOV) is associated with a moderate form of HFRS, with sporadic cases reported in Europe as well [5,6]. HNTV, responsible for the most severe form of HFRS, is primarily found in Asian countries and the Russian Far East [7].

The clinical presentation of HFRS varies depending on the causative serotype. The disease typically follows five distinct phases: a febrile phase, followed by hypotensive shock, oliguric, polyuric, and ultimately a convalescent phase. However, some of these phases may overlap or be absent altogether. Clinical manifestations can range from asymptomatic or mild forms to severe cases involving acute renal failure and haemorrhagic complications [4]. Common, but non-specific, symptoms include fever, headache, back or abdominal pain, and nausea or vomiting. Haemorrhagic manifestations may vary widely, from localised bleeding to severe, widespread haemorrhages [6,7,8].

The diagnosis of HFRS is primarily based on clinical presentation, which typically includes fever followed by abdominal pain, thrombocytopenia or haemorrhagic signs, and acute renal failure. Laboratory confirmation is usually achieved through serological testing, most commonly enzyme-linked immunosorbent assay (ELISA), detecting IgM and IgG antibodies against orthohantavirus antigens. In addition, immunoblot and immunofluorescence assays (IFAs) are widely employed for diagnostic purposes. RNA-based methods such as reverse transcription PCR (RT-PCR) are rarely used in routine clinical diagnostics, as viremia is typically short-lived, lasting approximately five to seven days [1,4]. However, molecular techniques are frequently applied in the study of host reservoirs and in the surveillance and molecular epidemiological characterisation of orthohantaviruses in rodent populations [3,4].

In the Republic of Kazakhstan, a Central Asian country, orthohantavirus infection was first identified in the West Kazakhstan region at the beginning of the new millennium in patients presenting with fever, abdominal pain, thrombocytopenia, and renal insufficiency [9,10]. Since then, annual cases of orthohantavirus infection have been reported in the region, with the exception of 2002, 2021, and 2022. According to the annual report on infectious and parasitic diseases in Kazakhstan (2000–2023), a total of 250 HFRS cases have been recorded, corresponding to an average annual incidence rate of 1.7 cases per 100,000 population. Following the initial detection of human cases, surveillance of natural rodent hosts was initiated. Over the years, more than 49,000 small mammals were trapped across the West Kazakhstan region and screened for orthohantavirus antigens using ELISA. Viral antigens were detected in 1.5% of small mammals, particularly in four northern districts: Terekti, Chingirlau, Borili, and Bayterek [10]. Two small-scale molecular studies identified the presence of *Orthohantavirus tulaense* (TULV) in individual samples from the West Kazakhstan and Almaty regions, respectively [11,12]. However, to date, no serotype classification of orthohantavirus infections in patients has been conducted using established serotyping methods such as indirect immunofluorescence assay (IIF), immunoblot, or PCR. To our knowledge, this study is the first to determine circulating orthohantavirus serotypes among patients with suspected HFRS in Kazakhstan.

## 2. Materials and Methods

### 2.1. Study Setting and Sample Collection

This cross-sectional study was conducted between April 2018 and December 2019 and involved individuals with suspected HFRS from two regions in the Republic of Kazakhstan. In West Kazakhstan, a confirmed endemic region, samples were collected from regional infectious disease hospitals. In Almaty city, a non-endemic region, samples were collected from infectious disease hospitals as well as from nephrology departments of central hospitals. A suspected case of HFRS was defined by the presence of symptoms such as fever, backache, abdominal pain, thrombocytopenia, and/or signs of haemorrhage and/or acute kidney failure. Individuals of all genders aged 18 years and older were eligible for inclusion. A standardised, paper-based, face-to-face questionnaire was administered to collect information on socio-demographic characteristics, living conditions, exposure to livestock and vector habitats, and clinical symptoms. Each participant was assigned a Hospital Identification Number (HIN), composed of a three-letter abbreviation of the city or region and a sequential three-digit number beginning with 001. On the first day of hospitalisation, serum, saliva, and urine samples (1st set) were collected. A second serum sample was obtained 10–14 days later.

### 2.2. ELISA Screening

All first and second serum samples were screened for the presence of orthohantavirus-specific IgG and IgM antibodies using a commercial ELISA kit (NovaTec Immunodiagnostica, Dietzenbach, Germany), following the manufacturer’s instructions. The results from the IgG and IgM ELISA screenings were grouped based on serological constellation patterns and subsequently interpreted as follows:
1^st^ and 2^nd^ serum IgM− and IgG−negativeNo infection1^st^ and/or 2^nd^ serum IgM+1^st^ and 2^nd^ serum IgG−IgM-positiveAcute infection1^st^ and 2^nd^ serum IgM−1^st^ and/or 2^nd^ serum IgG+IgG-positivePrevious infection or cross reactive1^st^ IgM+ and 2^nd^ IgG++Both IgM- and IgG-positiveAcute infection

### 2.3. Immunoblot Testing

IgM-positive, IgG-positive, and both IgM- and IgG-positive samples were further analysed using an IgG/IgM immunoblot assay (Mikrogen recomLine HantaPlus, Munich, Germany) to differentiate between orthohantavirus serotypes [13,14]. The immunoblot results were interpreted visually by assessing the intensity of antigen-specific bands relative to the cut-off band, categorised as follows:− (no reaction), ± (very low intensity), + (low intensity), ++ (high intensity), and +++ (very strong intensity).

### 2.4. RT-PCR

In patients with IgM+ serum samples, the corresponding urine and saliva specimens were analysed using RT-qPCR using a mix of pan-Hanta primers to detect a partial L-segment sequence (230 bp), using the Qiagen One Step RT-PCR mix complemented with EvaGreen (VWR International, Vienna, Austria) as PCR reagents [15].
**Primer Name****Sequence (5′–3′)****Direction**Pan-Hanta 1a-fwTgATgCATATTgTgTgCAgACForwardPan-Hanta 1b-fwTgATgCATACTgTgTgCAAACForwardPan-Hanta 1c-fwCAgTATgATgCATACTgTgTCCAAForwardPan-Hanta 1d-fwTgATgCCTATTgTgTTCAgACForwardPan-Hanta 1a-revCTTgCTCTgTTTTgAATCTCAReversePan-Hanta 1b-revCTTgCTCggTgTTgAATCgCAReversePan-Hanta 1c-revCCTgTTCTgTATTAAATCTCAReversePan-Hanta 1d-revCTTgTTCAgTCTTgAATCTCAReverse


### 2.5. Ethics Approval

This study was performed in accordance with the Kazakhstan local ethics committee at the Kazakh National Medical University in Almaty, Kazakhstan (opinion number 564-18), and the Ethics Committee of the Ludwig Maximilians University in Munich, Germany (opinion number 18-631). The blood, urine, and saliva samples were collected following the provision of written informed consent from all the participants.

### 2.6. Data Analysis

Statistical analysis was conducted using R version 4.4.2. Chi^2^ and Fisher’s exact tests were applied to assess associations between categorical variables, while the Mann–Whitney U test was used for continuous variables. Given the limited sample size, *p*-values were derived from 5000 bootstrap replications. Statistical significance was defined as α = 0.05.

## 3. Results

### 3.1. Sample Overview and ELISA-Based Screening Results

During the study period in 2018 and 2019, a total of 146 patients with suspected HFRS were enrolled from the West Kazakhstan region (one hospital) and Almaty City (three hospitals). After excluding patients with incomplete records, 139 cases (57 from West Kazakhstan, 82 from Almaty) were included in the final analysis. For these individuals, paired serum samples (first sample collected at admission and second sample collected 10 to 14 days later), along with urine, saliva, and a completed questionnaire, were available. All 139 patients were screened for orthohantavirus-specific antibodies using ELISA targeting IgM and IgG. Initial IgG serology testing of the paired serum samples revealed reactivity in 34 cases (24.5%) overall, including 24 of 57 cases (42.1%) in the West Kazakhstan region and 10 of 82 cases (12.2%) in Almaty City. Subsequent IgM ELISA testing identified two IgM+ cases in Almaty City, and five IgM+/IgG+ double-positive cases in the West Kazakhstan region, indicating acute infections (Figure 1, Appendix A).

### 3.2. Serotype Differentiation by Immunoblot

A succeeding immunoblot analysis demonstrated varying seroreactivity to known orthohantavirus strains. The immunoblot assay was performed on all ELISA IgM+ (*n* = 2), IgG+ (*n* = 34), and IgM/IgG double-positive (*n* = 5) serum samples (Table 1). Among the 34 ELISA IgG-positive sera, only 14 (41.2%) showed seroreactivity in the immunoblot assay. Of these, 12 samples exhibited high band intensity (++high-intensity band) specific to Puumala virus nucleocapsid protein (PuN), 1 sample showed weak reactivity (+/−very-low-intensity band) to Hantaan virus (HaN), and 1 to Dobrava virus (DobN). Among the two ELISA IgM-positive and five double-positive serum samples, six showed high immunoreactivity to PuN (++high-intensity band), with some also displaying weak cross-reactivity to Sin Nombre virus. One sample yielded inconclusive results, lacking definitive banding patterns.

### 3.3. Molecular Confirmation of Acute Cases, Demographic, and Clinical Features

All seven samples with serological evidence of acute orthohantavirus infection (IgM+ or IgM+/IgG+), comprising serum, urine, and saliva, were further tested for orthohantavirus RNA by RT-qPCR. Of these, five serum samples originated from patients in the West Kazakhstan region, specifically from the regional infectious disease hospital (samples: URA-007, URA-016, BUR-001, BUR-003, and BUR-024), and two samples were collected in Almaty City, one from the nephrology department (ALB-003) and one from the infectious disease hospital (ALM-028) (Table 1). The majority of patients (6/7, 85.7%) were male, with ages ranging from 28 to 62 years (28, 29, 32, 33, 34, 38, and 62 years). The analysis of living and housing conditions revealed that two patients resided in rural areas, while five lived in urban settings. Housing types included five apartments and two detached houses. The immediate environment surrounding the residences included dense vegetation (*n* = 3), proximity to agricultural or large grassy fields (*n* = 2), and one swampy area (*n* = 1).

Regarding potential exposure to reservoirs or vector habitats, three patients reported travelling into natural areas prior to symptom onset, with two of these excursions occurring near recognised endemic zones. None of the patients reported direct contact with wild animals, arthropod vectors (ticks or mosquitoes), or bites from small mammals. However, four patients had observed rodents in their environment. None reported known exposure to rodent excreta.

The most common clinical symptoms among ELISA IgM+ individuals were fever, headache, fatigue, back pain, and abdominal discomfort (Figure 2). Routine blood analyses revealed thrombocytopenia (platelet count < 180 g/L; reference range: 180–320 g/L) in all the IgM-positive patients, although no clinical signs of haemorrhage were observed (Figure 2, Appendix A). Four IgM-positive cases from the West Kazakhstan region were confirmed as HFRS using the ELISA IgM test from Vector Best (Russia), which is the regional diagnostic standard. One additional patient from this group was hospitalised with a provisional diagnosis of fever of unknown origin.

In Almaty City, one IgM-positive case from the infectious disease hospital was initially diagnosed with an enterovirus infection. Another IgM-positive case from the nephrology department was hospitalised for acute tubulointerstitial nephritis. This patient had recently undertaken a nature excursion near Almaty City and reported visual contact with rodents during the trip.

The results of the RT-qPCR screening revealed the existence of orthohantavirus RNA in three serum samples and three urine samples of a total of six patients (BUR-001 serum, BUR-003 serum, BUR-024 serum, URA-007 urine, URA-016 urine, and ALB-003 urine).

### 3.4. Clinical Profile of IgG-Positive Individuals

Patients who tested positive for IgG antibodies against orthohantavirus in the first and/or second serum sample exhibited a clinical profile characterised by common symptoms such as fever (97.1%), headache (76.5%), fatigue (73.5%), abdominal pain (50.0%), earache (23.5%), back pain (17.6%), arthralgia (14.7%), nasal congestion (14.7%), sore throat (11.8%), visual disturbances (8.8%), and lymphadenopathy (17.6%). The statistical analysis revealed that residing in a house was significantly associated with a higher likelihood of IgG seropositivity compared to living in an apartment (*p* < 0.01). A significant association was also observed between rural residency and IgG seropositivity (*p* < 0.004). Furthermore, the type of vegetation surrounding the residence appeared to influence infection risk, with environments such as agricultural fields, swamps, lakes, and forests associated with increased seropositivity (Appendix A).

## 4. Discussion

HFRS and HCPS caused by orthohantavirus infection continue to pose a significant public health concern. The highest number of reported human cases occurs in China, predominantly associated with the HNTV and SEOV serotypes, and in Russia, where infections are primarily linked to DOBV and PUUV. In Europe, the majority of cases are reported from Balkan countries, with PUUV and DOBV being the predominant circulating serotypes. A strong host–virus association exists between specific orthohantavirus species and their rodent reservoirs, suggesting a long-standing co-evolutionary relationship [16].

The present study represents the first systematic analysis of suspected HFRS cases in the Republic of Kazakhstan, a representative Central Asian country, comparing the endemic West Kazakhstan region with the non-endemic Almaty City. Previous investigations in West Kazakhstan have demonstrated orthohantavirus presence in both rodent host reservoirs and human cases. Routine registration of HFRS in this region began in 2000, with an incidence peak of 16 per 100,000 population documented in 2005 [10]. Although PUUV is presumed to be the dominant circulating serotype in West Kazakhstan, this has not yet been confirmed by molecular studies. Notably, our 2018–2019 rodent reservoir study detected TULV in one positive sample from the West Kazakhstan region (Figure 3) [11].

This finding raises several important questions. First, the pathogenic potential of TULV in humans remains poorly defined. Although TULV-associated HFRS cases have been reported in Europe, most infections are believed to go undiagnosed due to their typically mild or asymptomatic presentation [17,18]. Second, TULV shares close genetic and antigenic similarities with PUUV, which complicates differentiation by standard serological methods. Consequently, a proportion of the reported PUUV cases may in fact reflect misdiagnosed TULV infections [19].

Our study reports fever, headache, and fatigue as the most frequently observed clinical symptoms among orthohantavirus ELISA-positive individuals, consistent with previous findings [20,21]. However, these symptoms are non-specific and may lead to misdiagnosis, especially in non-endemic regions where standardised HFRS diagnostics are lacking. Additionally, mild or asymptomatic clinical courses are common in orthohantavirus infections, further complicating clinical recognition and timely diagnosis [22].

Several seroprevalence studies have been conducted among targeted population groups to better elucidate the routes of orthohantavirus infection [23,24]. In the present study, we focused on individuals classified as suspected HFRS cases from both endemic and non-endemic regions of Kazakhstan. Among the five confirmed positive cases from West Kazakhstan, where HFRS is endemic, we identified PUUV as the causative serotype through immunoblot and RT-qPCR. Notably, we also confirmed one PUUV-positive HFRS case in Almaty City, a region located more than 2000 km southeast of the known endemic areas. This patient was admitted to a nephrology department, suggesting that the infection occurred in a setting where HFRS is not typically suspected.

These findings yield several key insights: (I) HFRS cases may be underdiagnosed or misdiagnosed due to their non-specific clinical presentation; (II) awareness of HFRS remains limited in non-endemic areas; and (III) the emergence of a case in Almaty City may reflect increasing human mobility and economic activity, as the city functions as a major national and regional hub. This interpretation is supported by previous rodent surveillance studies conducted in 2018–2019 in Almaty City, which failed to detect orthohantavirus in any captured rodents, and the findings from the neighbouring Zhetisu region that identified only TULV, a serotype with uncertain pathogenicity in humans [11,12].

Importantly, our study successfully detected orthohantavirus RNA in both serum and urine samples from acute cases, suggesting that molecular diagnostics, including RT-qPCR on clinical samples such as saliva and urine, may provide a valuable tool for early and non-invasive diagnosis, particularly in severe cases [25,26]. Additionally, we observed significant differences in seropositivity rates between rural and urban residents. However, lifestyle factors such as leisure outdoor activities (e.g., nature trips and gardening), which are common in both West Kazakhstan and Almaty City, likely contribute to increased risk of exposure to infected rodent reservoirs or their excreta.

This study has several limitations that should be acknowledged. First, cross-reactivity among orthohantavirus serotypes and potential non-specific serological responses are the well-documented limitations of ELISA and immunoblot assays. While a focus reduction neutralisation test (FRNT) would serve as a confirmatory gold standard, it is currently unavailable in Kazakhstan. Second, a key limitation of this study is the inability to confirm the specific hantavirus species responsible for the detected infections through sequencing. Although RT-qPCR targeting a conserved region of the L segment successfully detected hantavirus RNA in several acute-phase samples, repeated attempts to sequence the amplified products were unsuccessful. This is likely attributable to low viral RNA concentrations and the potential degradation of clinical specimens collected under field conditions. Consequently, we were unable to distinguish between closely related hantavirus species such as Puumala virus (PUUV) and Tula virus (TULV), which are known to exhibit considerable serological cross-reactivity. While our serological and molecular data strongly suggest PUUV as the causative agent, definitive species identification will require future studies with improved sample quality, higher viral loads, and successful sequencing. Third, the relatively small sample size limits our ability to draw robust associations between socio-demographic or environmental factors and seropositivity. Thus, while we cannot definitively conclude that HFRS is circulating in Almaty City, the detection of a single PUUV-positive case underscores the need for further investigation. It is plausible that additional, undetected cases exist, and that rodent populations in urban centres like Almaty may eventually serve as viral reservoirs.

We, therefore, strongly recommend comprehensive follow-up studies targeting both human populations and rodent reservoirs in non-endemic regions such as Almaty City to better understand the epidemiology and potential expansion of orthohantavirus circulation in Kazakhstan.

## 5. Conclusions

HFRS and HCPS caused by orthohantavirus remain significant public health concerns, with confirmed endemic areas such as West Kazakhstan showing Puumala serotype circulation in both rodent reservoirs and human cases. A single HFRS case in non-endemic Almaty City suggests potential underdiagnosis due to non-specific symptoms, lack of diagnostic capacity, and increased human mobility. The study highlights the need for improved diagnostic methods, expanded surveillance in both endemic and non-endemic regions, and larger studies to assess true disease prevalence and transmission risks.

## Figures and Tables

**Figure 1 viruses-17-00925-f001:**
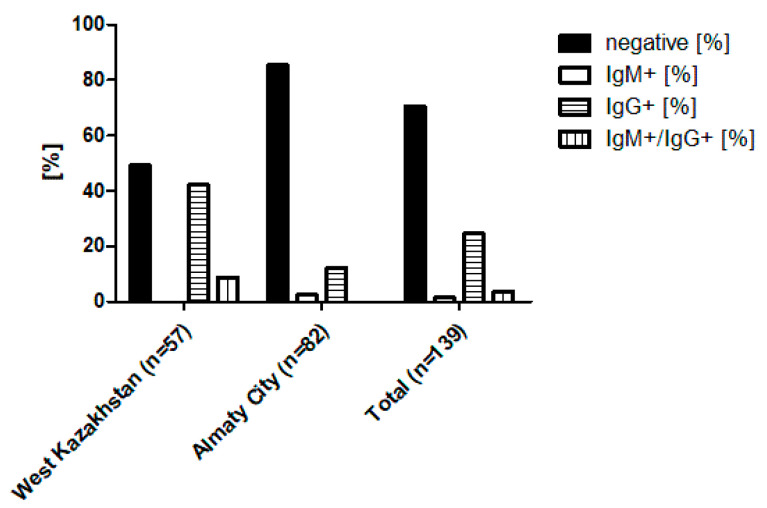
Percentage of ELISA IgM and IgG seropositivity for orthohantavirus in serum samples from suspected HFRS cases in the West Kazakhstan region (endemic area) and Almaty City (non-endemic area). ‘Total’ refers to the combined patient data from both regions.

**Figure 2 viruses-17-00925-f002:**
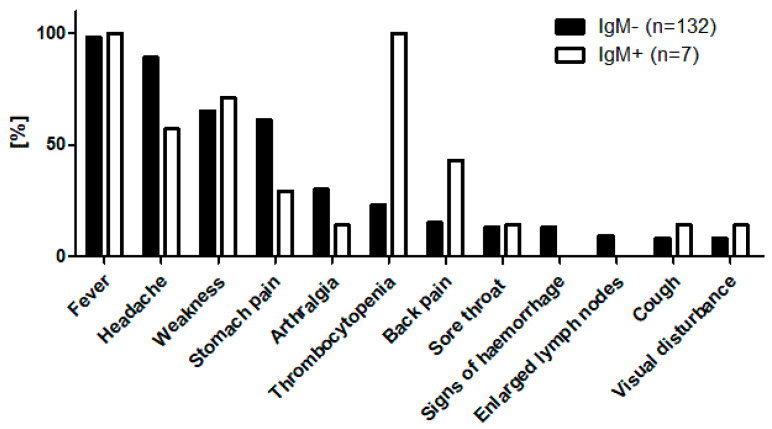
Percentage frequency of clinical signs among the tested subjects, stratified by ELISA IgM serostatus (IgM-positive vs. IgM-negative). The clinical signs listed are typical for patients presenting with HFRS.

**Figure 3 viruses-17-00925-f003:**
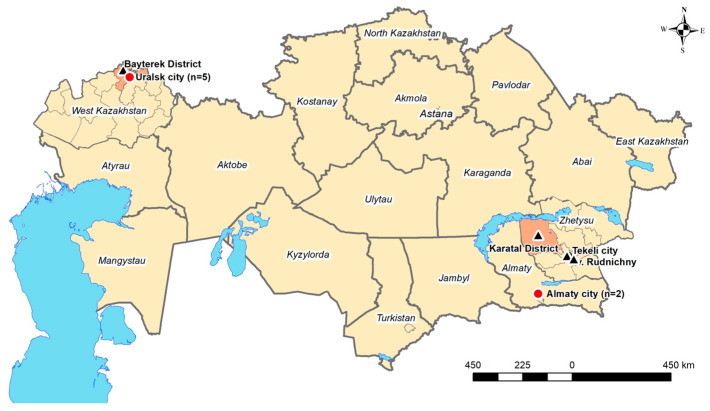
Geographical location of the sampling sites and positive results for orthohantavirus infection in humans and small mammals in the Republic of Kazakhstan. ● Positive human sample serology; **▲** positive in small mammals.

**Table 1 viruses-17-00925-t001:** Combined laboratory test results on the seven IgM+ samples collected in either the West Kazakhstan region or Almaty City (2018–2019) with their corresponding IgG levels, immunoblot results, RT-qPCR readout, and initial hospital diagnosis.

Sample ID	Age	Sex	ELISA	Immunoblot IgM Strip Intensity Rating	RT-qPCR(Positive Samples)	Initial Hospital Diagnosis
IgM1^st^ Serum	IgG2^nd^ Serum	PuN	HaN	DobN	SeoN	SinN
URA 007	38	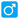	pos	pos	++	+/−	+/−	−	+/−	urine	HFRS
URA 016	28	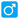	pos	pos	++	−	−	−	+/−	urine	HFRS
BUR 001	62	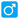	pos	pos	++	−	−	−	+/−	serum	HFRS
BUR 003	34	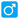	pos	pos	++	−	−	−	+/−	serum	Fever of unknown origin
BUR 024	33	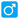	pos	pos	++	−	−	−	+/−	serum	HFRS
ALM 028	32	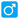	pos	neg	−	−	−	−	−	negative	Enterovirus infection
ALB 003	29	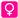	pos	neg	++	−	−	−	+/−	urine	Acute tubulointerstitial nephritis

Abbreviations: URA, BUR—Oral Regional Infectious Disease Hospital; ALM—Almaty Infectious Disease Hospital; ALB—Almaty Nephrology Department; ELISA—enzyme-linked immunosorbent assay; pos—positive in ELISA; neg—negative in ELISA; IgM—Immunoglobulin M; IgG—Immunoglobulin G; PuN—Puumala; HaN—Hantaan; DobN—Dobrava; SeoN—Seoul; SinN—Sin Nombre; HFRS—haemorrhagic fever with renal syndrome; ++high-intensity band (higher than cut-off); +/−very-low-intensity band (lower than the cut-off band); RT-qPCR—quantitative reverse transcription polymerase chain reaction.

## Data Availability

The data used and/or analysed during the current study are available from the corresponding author upon reasonable request.

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
