# Peer review of "Characterisation of Orthohantavirus Serotypes in Human Infections in Kazakhstan"

_viruses, 2025, doi:10.3390/v17070925_

Round 1
Reviewer 1 Report
Comments and Suggestions for Authors
In the paper by Tukhanova et al a serology/molecular biology analysis of the Kazhkhan region is described. The emergence of endemic infection is of interest in hantavirus research as many strains remain highly location-restricted to specific populations of rodents. Thus the authors conclusion that more surveillance is needed is justified and this paper is a contribution to the surveillance of emerging pathogens and thus valuable.
Figures - tables should be reformatted for better space use and clarity. A map of where positive samples were found would be a useful addition to make the paper more accessible. References and statistical analysis are appropriate. Sequence data is missing, which might have confirmed serology, and virus detection in rodents is also missing which detracts from the value of the paper. The authors also claim that the qPCR comfirms PUUV, however use a pan-hantavirus qPCR and state explicity in the text that PUUV and TULV cannot be distinguished serologically. Sample size is small.
English is generally good but could use screening by a native speaker.
Author Response
Figures - tables should be reformatted for better space use and clarity.
Our response: Thanks to the reviewer for pointing this out. We adapted it accordingly.
A map of where positive samples were found would be a useful addition to make the paper more accessible.
Our response: We thank the reviewer for this helpful suggestion. We agree that including a map enhances the clarity and geographic context of the study. Accordingly, we have added a map illustrating the study area in Kazakhstan, including the locations where human cases and positive samples were detected.
Sequence data is missing, which might have confirmed serology, and virus detection in rodents is also missing which detracts from the value of the paper.
Our response: Thank you for this insightful comment. We agree that sequencing data and virus detection in rodent reservoirs would have strengthened the conclusions of our study. Unfortunately, despite multiple attempts, sequencing of the PCR-positive samples was unsuccessful. This may be due to low viral RNA concentration or RNA degradation, which is a known limitation in field-collected clinical specimens.
Additionally, we agree that data on virus detection in rodent reservoirs significantly enhances the interpretation of human serological and molecular findings. In this context, we would like to highlight that a complementary rodent study was conducted in parallel with the present human investigation, during the same study period (2018–2019), and the results were published in Viruses, 2022, Tukhanova et.al,: Molecular Characterisation and Phylogeny of Tula Virus in Kazakhstan.
The authors also claim that the qPCR comfirms PUUV, however use a pan-hantavirus qPCR and state explicitly in the text that PUUV and TULV cannot be distinguished serologically. Sample size is small.
Our response: We appreciate the reviewer’s careful assessment. We agree that using a pan-hantavirus qPCR assay does not allow for definitive identification of PUUV, especially in the absence of sequence confirmation. In our original discussion, we acknowledged cross-reactivity in serology and the lack of FRNT testing as a limitation.
Regarding the sample size, we agree that it is relatively small. We initially calculated a required sample size of 380 participants to achieve sufficient statistical power for association analysis. However, we were unable to reach this target due to real-world challenges: a proportion of patients did not meet the inclusion criteria, and some declined participation. This limitation of the study is elaborated in the discussion part of the manuscript.

Reviewer 2 Report
Comments and Suggestions for Authors
This manuscript elucidates the significance of employing serological and molecular methodologies for the identification of "serotypes" of hantaviruses in circulation within Kazakhstan. The results have the potential to contribute novel data that could improve the surveillance of hantavirus in the country.
The following points are to be considered:
Line 52
Could you please confirm that the citation is correct? It may need a supplementary citation for European hantaviruses.
Line 59
Here appears number "4". Is this a citation?
Line 86
The concept of 'serotype definition' is introduced here. Please provide a definition, as the discriminatory capacity could differ depending on whether a total or truncated recombinant protein is used, for example.
TABLE 4
Too many decimals. For the "age" category, this decimal have not a biological significance
LINE 89
In the expression “…orthohantavirus infections among patients with suspected cases of HFRS. Could be “ …patients with suspected HFRS? Alternatively: …suspected HFRS cases?
LINE 112
I think you mean Mikrogen with a 'K'.
Are there any published references using this kit for the serological differentiation of Hantaviruses? To reinforce your results, it would be useful to mention any other studies that have used this kit for serotyping.
The test used cannot exclude an infection with the Hanta virus and/or Sandfly Fever Virus if the result is negative for recomline HantaPlus IgG, IgM. In particular, antibodies may not be present, or may not be present in detectable quantities, in the early phase of the infection. If clinical symptoms indicate an infection but the test results are negative and/or inconclusive, another sample should be taken and tested after two to three weeks (known as the 'diagnostic window'). In rare cases, no IgM or IgG antibodies are formed due to immunosuppression. Have the authors considered these possibilities in relation to the negative results?
LINE 117
The source of the primer used was missing. A citation or brief description of the design could be necessary. Which viral segment was amplified?
LINE 130
How were these statistical tests applied? What software was used?
LINE 255
The RT-PCR results are presented here. Was the amplified fragment sequenced? For endpoint RT-PCR, nucleotide sequencing was recommended to confirm the specificity of the obtained band.
Another issues:
1. Adding a map representing the study area would be helpful. Furthermore, the map could include the reported genotype/serotype.
2. I could not find citation number 23 in the manuscript, which is referred to in the Reference section.
3. The authors affirm that the positive cases were caused by the Puumala virus. Since Puumala and Tula are very similar, it is relevant to discuss whether these Puumala cases could be Tula virus infected cases. The Immunoblot IgG/M (MiKrogen) does not consider Tula virus infected cases.
Author Response
Line 52: Could you please confirm that the citation is correct? It may need a supplementary citation for European hantaviruses.
Our response: We thank the reviewer for pointing this out. We agree that additional references could enhance the context, particularly concerning European hantavirus distribution and characteristics. Therefore, we have added a supplementary citation to provide more comprehensive coverage of hantavirus epidemiology in Europe: [Vaheri, A.; Henttonen, H.; Voutilainen, L.; Mustonen, J.; Sironen, T.; Vapalahti, O. Hantavirus infections in Europe andtheir impact on public health.Rev. Med. Virol.2013,23, 35–49.]
Line 59: Here appears number "4". Is this a citation?
Our response: We thank the reviewer for their careful reading. Yes, the number “4” is a citation. We have reviewed the reference and confirmed its relevance to the statement.
Line 86: The concept of 'serotype definition' is introduced here. Please provide a definition, as the discriminatory capacity could differ depending on whether a total or truncated recombinant protein is used, for example.
Our response: Thank you for the remark. We want to highlight that we did not isolate actual virus from the patients but only RNA from the serum. To our understanding a “serotype definition” is based on antigen-antibody interaction of a virus isolate. Here we are discussing about “serotype classification” where the serotype is deducted from more indirect methods such as PCR or ELISA. To avoid any confusion, we rephrased the referring sentence to: However, to date, no serotype classification of orthohantavirus infections in patients has been conducted using established serotyping methods such as indirect immunofluorescence assay (IIF), PCR, or ELISA.
TABLE 4: Too many decimals. For the "age" category, this decimal have not a biological significance
Our response: We thank the reviewer for this observation. We agree that reporting age with excessive decimal precision is not meaningful. We have corrected this in the revised manuscript by rounding the age values appropriately.
LINE 89: In the expression “…orthohantavirus infections among patients with suspected cases of HFRS. Could be “ …patients with suspected HFRS? Alternatively: …suspected HFRS cases?
Our response: We thank the reviewer for this helpful suggestion. We agree that the phrasing can be improved for clarity and conciseness. We have revised the text to read: “…patients with suspected HFRS”, which we believe is the most direct and grammatically smooth option. The manuscript has been updated accordingly.
LINE 112: I think you mean Mikrogen with a 'K'.
Are there any published references using this kit for the serological differentiation of Hantaviruses? To reinforce your results, it would be useful to mention any other studies that have used this kit for serotyping.
The test used cannot exclude an infection with the Hanta virus and/or Sandfly Fever Virus if the result is negative for recomline HantaPlus IgG, IgM. In particular, antibodies may not be present, or may not be present in detectable quantities, in the early phase of the infection. If clinical symptoms indicate an infection but the test results are negative and/or inconclusive, another sample should be taken and tested after two to three weeks (known as the 'diagnostic window'). In rare cases, no IgM or IgG antibodies are formed due to immunosuppression. Have the authors considered these possibilities in relation to the negative results?
Our response: Thanks for pointing out the typo. We confirm that the correct spelling is Mikrogen (with a "K"). This has been corrected throughout the manuscript.
We appreciate the suggestion to strengthen our methodology section. We have reviewed the literature and added references to other studies that have used the Mikrogen recomLine HantaPlus IgG/IgM kit for hantavirus serological differentiation (see references Schultze et.al and Brockmann et.al added in the revised manuscript):
Schultze, D., Fierz, W., Matter, H. C., Bankoul, S., Niedrig, M., & Schmiedl, A. (2007). Cross-sectional survey on hantavirus seroprevalence in Canton St. Gallen, Switzerland. Swiss medical weekly, 137(1-2), 21–26.
Brockmann, J., Kleines, M., Ghaffari Laleh, N., Kather, J. N., Wied, S., Floege, J., & Braun, G. S. (2024). A simple clinical score to reduce unnecessary testing for Puumala hantavirus. PloS one, 19(5), e0304500.
LINE 117: The source of the primer used was missing. A citation or brief description of the design could be necessary. Which viral segment was amplified?
Our response: We thank the reviewer for highlighting this omission. In our study, we targeted a partial region of the L segment of the hantavirus genome (230 bp) using a real-time RT-PCR assay.
To detect this sequence, we used a Qiagen One Step RT-PCR mix. The primer mix consisted of multiple forward and reverse primers: pan Hanta 1a forward TgATgCATATTgTgTgCAgAC, 1b forward TgATgCATACTgTgTgCAAAC, 1c forward CAgTATgATgCATACTgTgTCCAA, 1d forward TgATgCCTATTgTgTTCAgAC and pan Hanta 1a reverse CTTgCTCTgTTTTgAATCTCA, 1b reverse CTTgCTCggTgTTgAATCgCA, 1c reverse CCTgTTCTgTATTAAATCTCA, 1d reverse CTTgTTCAgTCTTgAATCTCA. All primers were used at a final concentration of 0.125 µM each. EvaGreen (VWR International, Vienna, Austria) was used as the fluorescent dye for real-time detection as described Mossbrugger et.al.:
Mossbrugger, I., Felder, E., Gramsamer, B., & Wölfel, R. (2013). EvaGreen based real-time RT-PCR assay for broad-range detection of hantaviruses in the field. Journal of clinical virology: the official publication of the Pan American Society for Clinical Virology, 58(1), 334–335.
LINE 130: How were these statistical tests applied? What software was used?
Our response: The statistical analysis was performed using R software (version 4.4.2). Specifically, Chi2 and Fisher’s exact tests were used for categorical variables, and the Mann-Whitney U test was applied for continuous variables to assess associations between potential risk factors and seropositivity outcomes. Due to the limited sample size, we utilized 5,000 bootstrap replications to compute p-values, enhancing the robustness of the significance testing. A significance level of α = 0.05 was used throughout.
LINE 255: The RT-PCR results are presented here. Was the amplified fragment sequenced? For endpoint RT-PCR, nucleotide sequencing was recommended to confirm the specificity of the obtained band.
Our response: We thank the reviewer for this important observation. We fully agree that sequencing of the RT-PCR amplicon is essential for confirming the specificity of the detected fragment, particularly in endpoint PCR assays.
In our study, we attempted to sequence the amplified 230 bp fragment of the L segment. However, sequencing was not successful due to low viral RNA concentration and suboptimal amplicon quality, which likely reflects the low viral load typically found in clinical samples collected in the disease course.
Another issues:
- Adding a map representing the study area would be helpful. Furthermore, the map could include the reported genotype/serotype.
Our response: Thanks for this suggestion. We agree that including a map would enhance the clarity and geographic context of the study. Accordingly, we have added a map illustrating the study area in Kazakhstan, including the locations where human cases were detected.
- I could not find citation number 23 in the manuscript, which is referred to in the Reference section.
Our response: We thank the reviewer for identifying this oversight. Citation number 23 was inadvertently included in the reference list but not cited in the main text. We have now carefully reviewed the manuscript and removed the reference
- The authors affirm that the positive cases were caused by the Puumala virus. Since Puumala and Tula are very similar, it is relevant to discuss whether these Puumala cases could be Tula virus infected cases. The Immunoblot IgG/M (MiKrogen) does not consider Tula virus infected cases.
Our response: We thank the reviewer for this important and insightful comment. We fully agree that due to the high degree of serological cross-reactivity between Puumala virus (PUUV) and Tula virus (TULV), and given that the Mikrogen recomLine HantaPlus IgG/IgM immunoblot does not include TULV-specific antigens, it is not possible to definitively exclude TULV infection in the PUUV-positive cases detected in our study (line 235-245 in discussion part).
To differentiate PUUV from TULV infection with higher specificity, a focus reduction neutralization test (FRNT) would be the preferred method. However, this assay is currently not available in Kazakhstan, which further constrained our ability to confirm serotype specificity.

Reviewer 3 Report
Comments and Suggestions for Authors
In this manuscript, Tukhanova et al assess suspect hantavirus patient samples using ELISA and RT qPCR and subsequently connect a number of samples with Puumala virus infection. Through serology studies are an important aspect of virology, the results presented here are confusing and the manuscript is difficult to follow at times.
Major comments:
- The methods section is lacking sufficient detail.
- The data is confusing as presented and manuscript can be difficult to follow at times.
- Could the authors please include additional information regarding the virology of the patient samples (ex. qRT PCR and ELISA for each sample and virus tested).
The manuscript is difficult to follow at times and should be revised for clarity.
Author Response
The methods section is lacking sufficient detail.
Our response: We thank the reviewer for this comment and appreciate the opportunity to clarify. We would like to note that detailed descriptions of the study design, setting, inclusion criteria, sample collection, serological testing (ELISA and immunoblot), and molecular methods (RT-qPCR with specific primer sequences) are already provided in the Methods section.
The data is confusing as presented and manuscript can be difficult to follow at times.
Our response: We thank the reviewer for this valuable feedback. We understand the importance of clear and logical data presentation. In response, we have thoroughly revised the manuscript to improve clarity and added subheadings to improve structure and readability:
-Sample overview and ELISA-based screening results
-Serotype differentiation by Immunoblot
-Molecular confirmation of acute cases and demographic, clinical features
-Clinical profile of IgG positive individuals
Could the authors please include additional information regarding the virology of the patient samples (ex. qRT PCR and ELISA for each sample and virus tested).
Our response: We thank the reviewer for the suggestion. The detailed results of qRT-PCR and ELISA testing, including IgM and IgG results for each patient sample, are presented in the Results section of the manuscript. We have provided summary tables and descriptions that specify the test outcomes per sample where available.
To improve clarity, we have reviewed and enhanced the presentation of these data to ensure ease of interpretation by readers. If the reviewer prefers, we would be happy to include an additional supplementary table with individual sample-level data for further transparency.

Round 2
Reviewer 3 Report
Comments and Suggestions for Authors
Summary:
In the revised manuscript, the authors addressed some reviewer comments, however the manuscript remains difficult to follow as presented. The detection and analysis of hantaviruses and HFRS in Kazakhstan is warranted and these results have potential to fill gaps in knowledge but the presentation here should be thoroughly revised and restructured.
Major comments:
- Revisions to the text did clarify some reviewer comments, however this manuscript is still difficult to follow and should be revised for clarity.
- As presented, the tables are difficult to follow and do not appear to have been revised compared to the original submission. Could the authors please restructure tables to be more clear and concise? The authors should consider separating out some of the information presented in table 4 into supplemental tables or graphs.
- Table 1: Format is confusing and very difficult to follow. Capitalization and grammar are not consistent and it’s unclear what the authors are trying to present
- Table 3: Confusing as presented. Formatting is very inconsistent. For the IgM strip test, results are not defined in the table legend. The text does mention intensity bands in lines 114-120, however this is not linked to table 3 nor explained well in the methods section.
- Overall, table legends are lacking and need more detail.
- Methods section is still confusing as presented and must be revised for clarity.
Minor comments:
- Line 121-129 The authors should consider including the PCR primers used as a table or format as a list for clarity.
- Table 3. Did the authors compare the initial and follow up patient serum samples using the IgM strip test and RT-qPCR in addition to the ELISA data?
- Line 247-253. Since TULV and PUUV have high serological cross-reactivity, can the authors differentiate these samples by RT qPCR to confirm PUUV vs TULV?
Author Response
Comment 1: Revisions to the text did clarify some reviewer comments, however this manuscript is still difficult to follow and should be revised for clarity.
Our response 1: We thank the reviewer for taking the time to re-evaluate our manuscript and for their continued efforts to improve its clarity and quality. In response, we have undertaken a thorough revision of the entire text with a specific focus on improving clarity, flow, and readability. However, we are a little reluctant to drastically change the sequence of storytelling since it is important to us to highlight the increase in granularity of the analysis. From initial hospital/doctors’ diagnosis to serological to molecular biological testing. With each step we increase the level of diagnostic quality, however we also lose some samples along the way due to issues in quality. The final step, the sequencing, is not possible at all, what we discuss in the Discussion.
Comment 2: As presented, the tables are difficult to follow and do not appear to have been revised compared to the original submission. Could the authors please restructure tables to be more clear and concise? The authors should consider separating out some of the information presented in table 4 into supplemental tables or graphs.
Our response 2: We thank the reviewer for this helpful comment. To improve the clarity and readability of the manuscript, we have created figures out of the tables 1 and 2 and hope this change in presentation now meets the expectation.
Furthermore, we transferred former Table 4, which contained detailed laboratory and clinical information for individual IgM-positive cases, to the supplementary materials (Supplementary Table 3).
Comment 3: Table 1: Format is confusing and very difficult to follow. Capitalization and grammar are not consistent and it’s unclear what the authors are trying to present
Our response 3: We thank the reviewer for this remark. As outlined above, we made this table in a figure and transferred the original table into the supplement (Supplementary table 1), with a revision regarding clarity, formatting, and consistency.
Comment 4: Table 3: Confusing as presented. Formatting is very inconsistent. For the IgM strip test, results are not defined in the table legend. The text does mention intensity bands in lines 114-120, however this is not linked to table 3 nor explained well in the methods section.
Our response 4: Thank you for this remark. Table 3, now in the current version it is table 1, represents a pool of all laboratory test information on the seven patients of interest. We deliberately put that table at the end of the results part since it represents the summary of all findings and we think it nicely summarises how we came to our conclusions. To reference to this table we included reference throughout the results section where appropriate.
Comment 5: Overall, table legends are lacking and need more detail.
Our response 5: We thank the reviewer for this helpful observation. In response, we have revised and expanded all table legends to ensure they are clear, informative, and self-explanatory.
Comment 6: Methods section is still confusing as presented and must be revised for clarity.
Our response 6: Thank you for your valuable feedback regarding the Methods section. We appreciate your concern and understand the importance of clarity in presenting methodological details. We would like to clarify that the Methods section is structured into the following clearly defined subsections to enhance readability and flow:
- Study Setting and Sample Collection
- ELISA Screening
- Immunoblot Testing
- RT-PCR
Each subsection addresses specific components of the methodology to ensure a logical progression and facilitate understanding.
Comment 7: Line 121-129 The authors should consider including the PCR primers used as a table or format as a list for clarity.
Our response 7: We thank the reviewer for this helpful suggestion. In response, we have reformatted the PCR primer information (lines 121–129) into a clearly structured table to enhance readability and clarity. The table includes the primer names, sequences, and orientation (forward/reverse). This change ensures that the technical details are easier to locate and interpret.
Comment 8: Table 3. Did the authors compare the initial and follow up patient serum samples using the IgM strip test and RT-qPCR in addition to the ELISA data?
Our response 8: We thank the reviewer for this important question. In our study, both first and second serum samples were tested by ELISA for IgM and IgG. However, the immunoblot IgM strip test and RT-qPCR were performed only on the first serum sample from each patient that tested positive for IgM and/or IgG by ELISA. This approach was taken because immunoblot and PCR assays were used as confirmatory or complementary tools following ELISA screening.
Comment 9: Line 247-253. Since TULV and PUUV have high serological cross-reactivity, can the authors differentiate these samples by RT qPCR to confirm PUUV vs TULV?
Our response 9: We thank the reviewer for this insightful comment. We fully agree that due to the high serological cross-reactivity between Tula virus (TULV) and Puumala virus (PUUV), serological methods alone cannot definitively distinguish between infections with these two viruses.
While our pan-hantavirus RT-qPCR assay successfully detected hantavirus RNA in a subset of acute-phase samples, the assay targets a conserved region of the L segment and is not species-specific. As a result, it cannot differentiate between PUUV and TULV without follow-up sequencing of the amplicon.
Unfortunately, despite repeated attempts, we were unable to obtain sequence data from these RT-qPCR-positive samples, likely due to low RNA concentrations or degradation. This limitation is now more clearly acknowledged in the revised Discussion section.
We agree that virus-specific PCR assays or sequence-based identification will be essential in future studies to definitively differentiate PUUV and TULV in clinical samples.
